# InstantAS: Minimum Coverage Sampling for Arbitrary-Size Image Generation

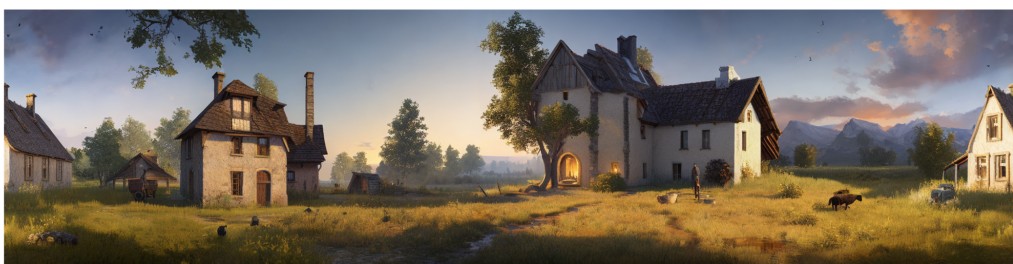

Some charming houses in the countryside, by jakub rozalski
896×3560   Sampling Time:12.3s

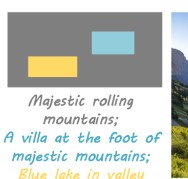 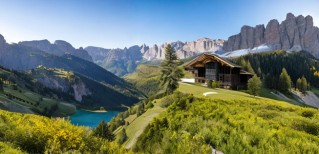

Majestic rolling mountains;
A villa at the foot of majestic mountains;
Blue lake in valley

523×1209   Sampling Time:5.1s

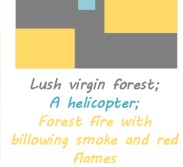 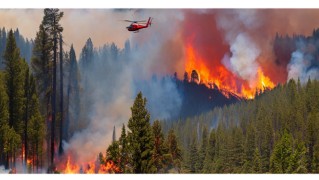

Lush virgin forest;
A helicopter;
Forest fire with billowing smoke and red flames

523×1209   Sampling Time:5.0s

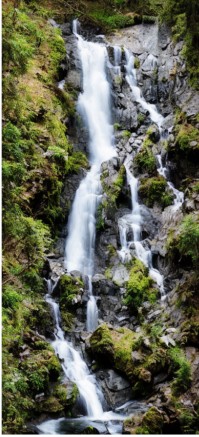

A cascading waterfall
2072×1040
Sampling Time: 10.9s

Figure 1: InstantAS can generate arbitrary-size image, such as horizontal panorama or vertical panorama. The generation speed of InstantAS is nearly four times that of commonly used MultiDiffusion [3] while maintaining generation quality. InstantAS can also apply different semantic information to different parts of the image during sampling and achieve seamless blending.

## ABSTRACT

In recent years, diffusion models have dominated the field of image generation with their outstanding generation quality. However, pre-trained large-scale diffusion models are generally trained using fixed-size images, and fail to maintain their performance at different aspect ratios. Existing methods for generating arbitrary-size images based on diffusion models face several issues, including the requirement for extensive finetuning or training, sluggish sampling speed, and noticeable edge artifacts. This paper presents the InstantAS method for arbitrary-size image generation. This method performs non-overlapping minimum coverage segmentation on the target image, minimizing the generation of redundant information and significantly improving sampling speed. To maintain the consistency of the generated image, we also proposed the Inter-Domain Distribution Bridging method to integrate the distribution of the entire image and suppress the separation of diffusion paths in different regions of the image. Furthermore, we propose the dynamic semantic guided cross-attention method, allowing for the control of different regions using different semantics. InstantAS can be applied to nearly any existing pre-trained Text-to-Image diffusion model. Experimental results show that InstantAS has better fusion capabilities compared to previous arbitrary-size image generation methods and is far ahead in sampling speed compared to them.

## CCS CONCEPTS

• **Computing methodologies** → **Computer vision**; *Image processing*; Machine learning.

## KEYWORDS

Image Generation, Diffusion Models, Fast Sampling, Training-Free

## 1 INTRODUCTION

In the field of image generation, diffusion models [5, 10, 11, 21, 26, 30, 31, 40] have achieved remarkable accomplishments in recent years. In terms of text-to-image generation models [4, 20, 25, 37, 38], many models that generate high-quality images with simplicity and usability have become popular. Existing text-to-image generation models are mostly trained on datasets of fixed sizes, leading to poor generalization for images of different sizes. Applications such as book covers, illustrations, and posters require flexibility in image sizes. Directly scaling images can compromise their coherence, while stitching images together can result in noticeable seams. Therefore, we need a method that can generate images of any size based on constrained pre-trained diffusion model.

Existing works have addressed this issue through various methods. Any-size-diffusion [41] and SDXL [23] fine-tune the model using various images of different sizes, achieving outstanding generation results. However, considering the model's parameter volume, this requires a large amount of training data and significant computational resources. ElasticDiffusion [8] decouples global and local generation and extracts the proportionate region from its generated results matching the target image size, then enlarges it to the target size using the proposed resampling technique. This method takes into account global information, but the large number of samples makes it very slow. ScaleCrafter [9] proposes a simple yet effective reexpansion that can dynamically adjust the convolution receptive field during the inference process, achieving high-quality super-resolution image generation. Some recent works [14, 19, 35, 36] adopt the MultiDiffusion [3] panoramic image generation method, which involves splitting large-sized images into different regions for separate generations, achieving excellent generation results. However, in these methods, the individually generated regions overlap, requiring a significant number of separately generated regions to be combined to form the target image during the generation process. This results in an exponential increase in the generation time as the region of the image grows.

In this paper, we propose the InstantAS method, which can generate images of any size at extremely high speed. To achieve this objective, we introduce the non-overlapping minimum coverage sampling method. This approach involves decomposing the large target image into smaller regions capable of independent generation. Compared to the large overlap of sampling areas in previous works [3, 14, 36], we require these regions to collectively form a precise cover of the large target image without any overlap. This meticulous segmentation minimizes the generated content compared to methods necessitating extensive overlaps and reduces the number of steps necessary in the sampling phase. Simply sampling images in block units may lead to gradual dispersion of probability flows among different blocks during the diffusion process, thereby creating distinct boundaries between images. To address this issue, we propose an inter-domain distribution bridging method, which harmoniously integrates various sampling regions during the generation phase to facilitate cohesive guidance for multiple score predictions. In addition, we further explored a more refined control method by guiding different regions of the image with different prompts. Specifically, we proposed the dynamic semantic guided cross-attention method, which dynamically allocates semantic information to different sampling regions under different diffusion steps by splitting the guiding parameters without the need for a classifier, achieving a more precise control.

We conducted extensive qualitative and quantitative comparative experiments on InstantAS to validate its superiority. The experimental results demonstrate that InstantAS outperforms state-of-the-art methods and is much faster than previous works [3, 8, 26] in generating arbitrary-size images. Overall, this paper makes the following three contributions:

- We propose a training-free arbitrary-size image sampling method that significantly outperforms all existing methods in terms of sampling speed while maintaining generation quality.

- We introduce a region-controlled generation method that allows for the generation of different regions within an arbitrary-size image using distinct semantic information, building upon our proposed fast sampling method.
- We demonstrate the outstanding capabilities of our method through analysis and extensive comparative experiments.

## 2 RELATED WORK

### 2.1 Conditional Diffusion Models

Conditional diffusion models has emerged as a powerful tool for controlling the synthesis process in generative tasks. Currently, conditional generation with diffusion models primarily falls into two forms: Classifier Guidance [5] and Classifier-Free [11]. Classifier Guidance uses an additional network to measure the degree of match between the intermediate results of the generation process and the conditions, and use its gradients to modify the generated results. Classifier-Free incorporates conditions into the generation process at the beginning of training in diffusion models.

ControlNet [40] utilizes the encoder of the U-net [27] in the pre-trained diffusion model to encode and input conditions, training with paired data to achieve outstanding results under various control conditions. T2I-Adapter [20] adjusts multiple conditions to a unified form through a pre-trained adapter, achieving multi-condition control in text-to-image generation without modifying the network. Some recent works [12, 17, 28, 39] focus on personalized content generation, enabling precise control over generated scenes, clothing, characters, and other content at minimal cost. In this paper, we propose a region-controlled method based on overlapping minimum coverage sampling, allowing to impose different semantic controls on different regions in images of arbitrary sizes.

### 2.2 Arbitrary Size Image Generation For Diffusion Models

Diffusion models [1, 5, 10, 11, 30, 31, 40] are an emerging class of generative models that progressively transform noise into structured data, positioned as an alternative to GANs [2, 6, 7, 13, 15, 16, 42] and VAEs [18, 33, 34]. Diffusion models boast exceptional generation quality, but their unique generation process results in significant training costs. Existing pre-trained diffusion models are typically built upon fixed image size datasets, leading to suboptimal performance when generating images of different resolutions. Consequently, arbitrary-size image generation methods based on pre-trained diffusion models have garnered widespread attention.

Multidiffusion [3] employs diffusion path merging, segmenting the target image of arbitrary size into fixed-size regions for generation. However, this approach necessitates extensive overlapping of the sampled regions, resulting in significant information redundancy and a substantial reduction in generation speed. ElasticDiffusion [8] extracts a proportionally scaled-down region from a fixed-size image and then upscale it to the target size, ensuring global information consistency. However, extensive upsampling also diminishes generation speed and leads to inferior generation quality for images with large aspect ratios. SyncDiffusion [19], building upon Multidiffusion's method, introduces optimization techniques to enhance image consistency. ScaleCrafter [9] proposes a simple rescaling method that dynamically adjusts the receptive

**Figure 2: The overall process of InstantAS. First, the noise image of the target size is divided into several sampling areas of different sizes according to the minimum coverage sampling method we proposed. Subsequently, these regions are filled respectively and then input into U-net, where dynamic semantic guided cross-attention is calculated with the corresponding prompts from different constrained regions. Finally, these sampling areas are assembled, the initial value of the next sampling is generated, and the sampling area is re-divided according to the inter-domain distribution bridging method.**

field of convolutions during inference. In contrast to these methods, InstantAS proposed in this paper significantly reduces information redundancy through non-overlapping minimum coverage sampling, achieving sampling speeds surpassing all existing arbitrary-size image generation methods while maintaining image quality.

## 3 INSTANTAS

The goal of this paper is to use a pre-trained diffusion model for rapid sampling of larger images with non-uniform aspect ratios while maintaining image consistency. To address this issue, we propose a non-overlapping minimum coverage sampling method. Specifically, for a target size of $H \times W$, we decompose the noise of the target size into multiple non-overlapping regions and sample them separately. To ensure consistency and seamlessness in the generated images, we also propose an inter-domain distribution bridging method to integrate the distribution differences between different regions. In addition, we introduce a dynamic semantic guidance intensity allocation method, which allows different semantic information to be applied to different regions during the generation process and naturally blend them, thereby enhancing the control precision in the sampling process. The complete method is shown in Figure 2.

### 3.1 Non-Overlapping Minimum Coverage Sampling

Some previous works[3, 14, 36] samples large-sized images by dividing them into different regions. However, in order to ensure the exchange of information between different sampling regions and avoid obvious boundaries between them, these methods generally

have large overlapping areas between different sampling regions. This result in neighboring regions being very close to each other, so to sample large-sized images, they must be divided into a large number of sampling regions. For example, MultiDiffusion[3] uses sampling regions with an interval of 8. For a panoramic image with a target size of $512 \times 2048$, it needs to generate $2048/8 = 256$ sampling regions. In these works, at least $87.5\%$ of adjacent sampling regions are required to overlap. However, to achieve a smooth transition between different areas, a large number of overlapping regions need to be sampled and then the average calculated. These repeated samplings create a large amount of information redundancy, thus greatly reducing the sampling speed.

To address this issue, we propose the Non-Overlapping Minimum Coverage Sampling method, as illustrated in the upper half of Figure 3. Specifically, let the target image be denoted as $\mathbb{I}$ with dimensions $H \times W$. As the current diffusion models commonly default to generating square images with a side length denoted as $L$, for simplicity, we set the size of each sampling region $\mathbb{R}_i$ as a square with a side length of $L$. Subsequently, the sampling regions form the minimum coverage on the target noisy image:

$$\{\mathbb{R}_i\}_{i=0}^{m} = Min\_Coverage\{\mathbb{I}, L\} \qquad (1)$$

For the part of the target image edge that is less than one sampling region length, we only allow temporary overlapping of two sampling regions in this area to meet the size of the target image. However, in the processing after each sampling step, we will remove the overlapping part used to fill the edge.

After segmenting the target image using our non-overlapping minimum coverage sampling method, the number of sampling regions obtained is much smaller than in previous works[3, 14, 36].

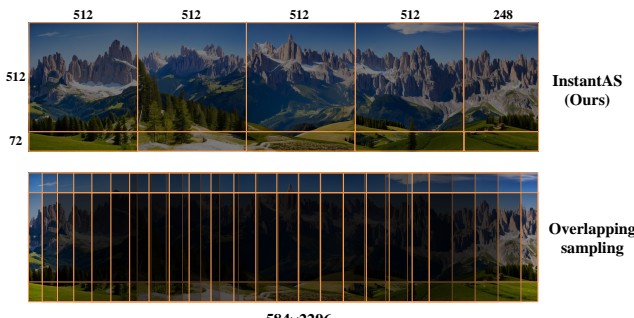

**Figure 3: On a** $584 \times 2296$ **image, we partition the sampling regions using non-overlapping minimum coverage sampling and overlapping sampling methods. Clearly, the area that needs to be sampled using InstantAS is far less than that required for overlapping sampling.**

For example, for simplicity, when using Stable Diffusion 2.0 [26] trained to generate $512 \times 512$ images to produce $512 \times 2048$ images, InstantAS requires sampling of $Ceil(2048/512) \times Ceil(512/512) = 4$ regions at each step, while the previous works would need to divide at least $2048/[(1 - 0.875) \times 512] = 32$ regions. InstantAS innovates the sampling method, eliminates the overlapping redundant information generated in the process of sampling images of any size, and greatly improves the sampling speed. Moreover, the original sampling results without averaging processing avoid the information confusion caused by multiple samplings, which improves the quality of the generated results to a certain extent, which will be detailed and compared in the Section 4.

## 3.2 Inter-Domain Distribution Bridging

If the sampling steps are only conducted in non-overlapping regions, the lack of information transfer between regions will cause them to gradually differentiate in the flow of the diffusion model's ODEs, leading to generated results in different regions being unrelated and showing obvious stitching traces. To address this issue, we propose the Inter-Domain Distribution Bridging method. In this method, we denote the distribution in the $t$-th sampling space of the diffusion model generation process as $P_t(\mathbf{x}_t|\mathbf{c})$, where $\mathbf{c}$ represents the prompt. The coordinates of two adjacent sampling regions in this distribution are denoted as $\mathbf{x}_t^{\mathbb{R}_i}$ and $\mathbf{x}_t^{\mathbb{R}_{i+1}}$. Since the initial value $\mathbf{x}_T$ of the entire target image is sampled from the standard normal distribution $\mathcal{N}(0, \mathbf{I})$ with covariance 0, if there is no information transfer between $\mathbf{x}_t^{\mathbb{R}_i}$ and $\mathbf{x}_t^{\mathbb{R}_{i+1}}$ from the beginning to the end, they are unlikely to converge to the same distribution. This discrepancy will eventually be reflected in the target image space $P_0(\mathbf{x}_0|\mathbf{c})$ with significant differences, as illustrated in Figure 4. Therefore, we realign the two coordinates to compensate for the differentiation that may occur in the next sampling:

$$\mathbb{R}_i \leftarrow Reorganize(\mathbb{R}_i, \mathbb{R}_{i+1}, \gamma) \quad (2)$$

Specifically, at the completion of the sampling in the $t$-th step, we select two adjacent sampling regions and choose a portion between them based on the fixed size of the sampling regions to form the sampling region for step $t - 1$. In our experiments, we choose

$\gamma$ from region $\mathbf{x}_t^{\mathbb{R}_i}$ and $1 - \gamma$ from region $\mathbf{x}_t^{\mathbb{R}_{i+1}}$, which form region $\mathbf{x}_{t-1}^{\mathbb{R}_i}$ after reorganization. $\gamma$, as a hyperparameter, is used to control the proportion of two adjacent images in a single reorganization. It is important to note that for incomplete sampling regions at the edges, we first fill them inwards, and after one sampling step is completed, we remove the excess parts. In addition, for sampling regions at the edge, we consider their neighboring regions as the starting sampling regions, ensuring that the number of sampling regions remains constant after each reorganization.

Figure 4(b) illustrates the principle of the inter-domain distribution bridging method. In practice, we arrange large images to form a ring by connecting them end to end, both from top to bottom and from left to right. In this way, the distribution bridging in each sampling step can be viewed as a cyclic shift downwards or backwards across all sampling regions currently involved. We control the length of the shift to ensure that each sampling region can cover a portion of the two sampling regions from the previous step. We found that choosing either a horizontal or vertical shift, but not both, within a single sampling step of the diffusion model yields satisfactory generative results. Therefore, we adopt an alternating shifting strategy: if a horizontal shift is used in the current sampling step, a vertical shift will be applied in the next step.

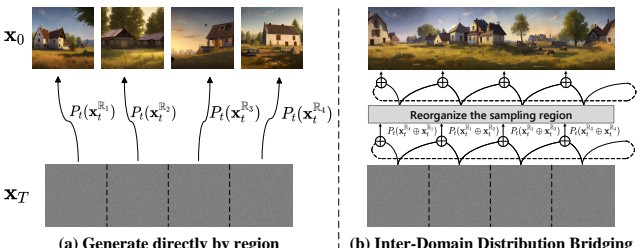

(a) Generate directly by region     (b) Inter-Domain Distribution Bridging

**Figure 4: Inter-Domain Distribution Bridging method can ensure the consistency of images. (a) Each sampling region is generated individually. Their distribution distance will grow increasingly larger. (b) The inter-domain distribution bridging method reorganizes the sampling regions to close in on the directions of same probability flows.**

## 3.3 Dynamic Semantic Guided Cross-Attention

Building upon our proposed non-overlapping rapid sampling method, we delved further into more refined control mechanisms, specifically, guiding the generation of different image regions with distinct semantic information. To achieve this objective, we introduce the Dynamic Semantic Guided Cross-Attention method, which allows for the dynamic distribution of semantic information to each sampling region throughout the generation process.

Before the generation starts, we have $k$ different guiding prompts: $\{\mathbf{c}^i\}_{i=1}^k$, each of which corresponds to $k$ different and non-overlapping regions $\{\mathbb{A}^i\}_{i=1}^k$ in image $\mathbb{I}$:

$$\mathbb{A}_t^i \Leftarrow P_t(\mathbf{x}_t^{\mathbb{A}^i}|\mathbf{c}^i) \quad (3)$$

During the generation process, since the sampling regions bridging the inter-domain distribution change at each step, we consider

$k$ different semantic guidance regions in the target-sized image and their corresponding semantics as the basis of the image. Each sampling region $\mathbb{R}_i$ intersects with some semantic guidance regions $\{\mathbb{A}^j\} \subseteq \{\mathbb{A}^i\}_{i=1}^k$, $j \in I_i$, and these semantic guidance regions precisely form an internal partition of the sampling region:

$$\mathbb{R}_i = \bigcup_{j \in I_i} (\mathbb{R}_i \cap \mathbb{A}^j) \qquad (4)$$

where $I_i$ represents the index set of those semantic guidance regions that intersect with $\mathbb{R}_i$. We modify the cross-attention module, which integrates semantic information with the generated image, by employing prompts within the intersection of each sampling region $\mathbb{R}_i$ and the semantically-guided regions $\{\mathbb{A}^j\}$ to compute cross-attention. This dynamically assigned cross-attention enables region-specific control of semantic information:

$$\text{Attn}_{\mathbb{R}_i} \leftarrow \underset{j \in I_i}{\text{Assemble}}\left( Softmax\left( \frac{(Q^{\mathbb{R}_i} \cap \mathbb{A}^j) \cdot K^{c_j}}{\sqrt{d}} \right) \cdot V^{c_j} \right) \qquad (5)$$

The modified attention map is input into U-Net for generation. After all sampling areas are generated, we remove some temporarily filled areas in the attention map and splice all sampling areas into the size of the original image for the next sampling, as shown in Figure 2. Complete sampling algorithm as shown in Algorithm 1.

---

**Algorithm 1** InstantAS Sampling Process

---

**Input:**

  $\mathbf{x}_T \sim \mathcal{N}(0, \mathbf{I})$           ▷ Noise at target size $H \times W$

  $\epsilon_\theta$           ▷ Pre-trained diffusion model at $L \times L$

  $\{\mathbf{c}^j\}_{j=1}^k$ and $\{\mathbb{A}^j\}_{j=1}^k$    ▷ Prompts and corresponding regions

1:  $\{\mathbb{R}_i\}_{i=0}^m = Min\_Coverage\{\mathbf{x}_t, L\}$      ▷ Eq.1

2:  **for** $t = T$ to $1$ **do**

3:     **for** $i = 0$ to $m$ **do**

4:        $\text{Attn}_{\mathbb{R}_i} \leftarrow \underset{j \in I_i}{\text{Assemble}}\left( Softmax\left( \frac{(Q^{\mathbb{R}_i} \cap \mathbb{A}^j) \cdot K^{c_j}}{\sqrt{d}} \right) \cdot V^{c_j} \right)$

5:        $\mathbf{x}_{t-1}^{\mathbb{R}_i} \leftarrow \epsilon_\theta(\mathbf{x}_t^{\mathbb{R}_i}, \text{Attn}_{\mathbb{R}_i}, t)$

6:     **end for**

7:     $\mathbb{R}_i^{(t-1)} \leftarrow Reorganize(\mathbb{R}_i^{(t)}, \mathbb{R}_{i+1}^{(t)}, \gamma)$    ▷ Eq.2

8:  **end for**

9:  $\mathbf{x}_0 = Min\_Coverage^{-1}\{[\mathbf{x}_0^{\mathbb{R}_0}, ..., \mathbf{x}_0^{\mathbb{R}_m}], L\}$

10:  return $\mathbf{x}_0$

---

## 4 EXPERIMENTS

*Evaluation Metrics.* In order to quantitatively evaluate the experimental results, we followed previous text-to-image generation work and used widely recognized and utilized metrics, FID (Fréchet Inception Distance) [22] and CLIP-Score, as evaluation indicators. FID utilizes the Inception v3 [32] image classification model to extract features and compute the similarity between two sets of images, used to measure image diversity and quality. In our experiment, we used the base generation model Stable Diffusion 2.0 to generate a set of $512 \times 512$ images based on a fixed prompt, then randomly cropped the large-size images generated by various comparative methods to obtain another set of $512 \times 512$ images, for FID calculation. We employed two CLIP-based evaluation metrics: (1) text-to-image CLIP score (CLIP-S) [24] encodes images and

prompts into the same latent space to calculate their cosine similarity, measuring the matching degree between generated images and prompts. (2) CLIP aesthetic (CLIP-A) [29] uses a linear estimator at the top of CLIP to obtain aesthetic indicators of the images. Furthermore, we introduced a very important metric: Sampling Speed. This metric is used to measure the efficiency of sampling methods and test whether they can maintain reasonable sampling times when increasing content generation.

*Baselines.* We compared InstantAS with five previous works: Stable Diffusion [26], MultiDiffusion [3], ElasticDiffusion [8], SyncDiffusion [19] and ScaleCrafter [9]. We use Stable Diffusion to sample directly on the noise of the target size. MultiDiffusion divides the image into overlapping small regions and samples them separately, then combines them to obtain the output. ElasticDiffusion starts from a small area in a fixed-size image to generate an image of target size. SyncDiffusion, based on the MultiDiffusion panoramic image generation method, proposes optimization strategies to make the image more coherent. ScaleCrafter proposed a simple and effective re-expansion method that can dynamically adjust the receptive field of the convolution during the inference process.

*Implementation Details.* InstantAS does not require any additional training. In the experiments, for fair comparison, all methods used the unified pre-trained text-to-image diffusion model Stable Diffusion 2.0 and the DDIM sampling method as the base, with diffusion steps set to 50 and guidance scale set to 7.5. For the Dynamic Semantic Guided Cross-Attention module, we set $\gamma$ to 0.015. All other settings were kept consistent. Additionally, all experiments were conducted on the same RTX 4090 GPU.

### 4.1 Qualitative Comparison

In this section, we conducted visual comparisons of all methods. We selected four different sizes within $2048 \times 2048$. The generation results are as shown in the Figure 5 and the Figure 6.

Figure 5 shows the results of six methods for generating horizontal long images. We present four sets of different sizes paired with prompts. Figure 6 shows the results of six methods on the generation of vertical long images. We still use randomly chosen image sizes within a reasonable range.

*Empty borders.* MultiDiffusion excels in both semantic restoration ability and generation results. However, for images whose size is not a multiple of 512, empty borders often appear to the right and below, affecting the quality of the generation. SyncDiffusion uses the same arbitrary-sized image generation method as MultiDiffusion, so the results also have similar content-free edges.

*Generation quality.* Stable Diffusion directly generates images of the target size, but it lacks the ability to generalize from the trained $512 \times 512$ size to larger sizes, resulting in repeated stacking of certain elements and poor generation quality. ElasticDiffusion performs poorly, producing blurry images with obvious seams. ScaleCrafter's generated results are visually impressive, but there are still repeating elements that are similar to each other.

In contrast, InstantAS performs excellently in all three experiments, showing high generation quality while perfectly matching

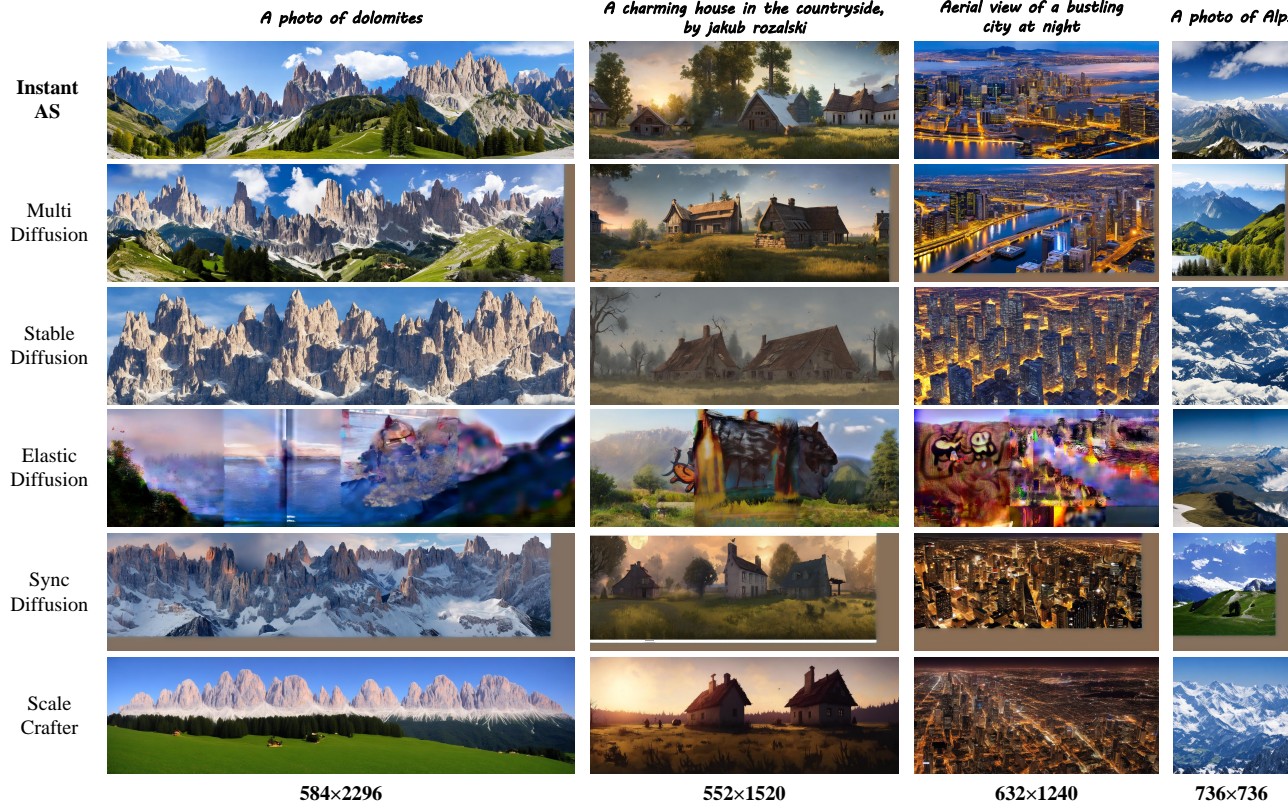

**Figure 5: The generation results of the six methods under four sets of different prompts and image size settings. For ease of display, we scaled the heights of the generated results for different sizes to compare.**

**Table 1: Quantitative comparisons. We performed a quantitative comparison of the following three metrics at four different dimensions: FID, CLIP-Score, CLIP-Aesthetic. We used 20 different prompts and sampled each size 2000 times.**

| Methods | 512 × 1024 | | | 512 × 2048 | | | 1024 × 512 | | | 2048 × 512 | | |
|---|---|---|---|---|---|---|---|---|---|---|---|---|
| | FID↓ | CLIP-S↑ | CLIP-A↑ | FID↓ | CLIP-S↑ | CLIP-A↑ | FID↓ | CLIP-S↑ | CLIP-A↑ | FID↓ | CLIP-S↑ | CLIP-A↑ |
| MultiDiffusion | 23.62 | 0.25 | 6.02 | 25.43 | **0.25** | 6.29 | 39.92 | 0.23 | 6.19 | 42.95 | 0.23 | 5.38 |
| StableDiffusion | 22.34 | 0.25 | 5.28 | 24.98 | 0.22 | 5.97 | **37.03** | 0.20 | 5.53 | 39.84 | 0.20 | 5.44 |
| ElasticDiffusion | 61.20 | 0.03 | 2.29 | 63.38 | 0.04 | 2.71 | 81.67 | 0.04 | 2.15 | 67.11 | 0.02 | 1.93 |
| SyncDiffusion | **20.19** | 0.24 | 5.72 | 25.17 | 0.24 | 6.02 | 38.93 | 0.22 | 6.02 | 40.18 | **0.25** | 6.09 |
| ScaleCrafter | 21.36 | **0.26** | 5.21 | 25.10 | **0.25** | 5.98 | 37.21 | 0.23 | 5.87 | 39.92 | 0.24 | 5.59 |
| **InstantAS** | 20.58 | **0.26** | **6.39** | **23.19** | 0.24 | **6.42** | 37.69 | **0.24** | **6.22** | **37.19** | 0.25 | **6.34** |

the target size. In conclusion, InstantAS demonstrates superior performance in image generation tasks of various sizes and proportions compared to other methods.

## 4.2 Quantitative Comparison

*Generate Quality Comparison.* In this section, we conducted quantitative experimental comparisons of the six methods. Specifically, for each method, we conducted sampling experiments using the same parameter settings. To facilitate metric calculations, we fixed four different image sizes: 512 × 1024, 512 × 2048, 1024 × 512, 2048 × 512. This allowed easy integration of CLIP and Inception v3. For

each size, we performed 2, 000 samplings for all methods and displayed the average scores of the sampling results, as shown in Table 1. MultiDiffusion performs well on both the FID metric and the two metrics calculated through CLIP, indicating excellent image quality and text consistency. However, because we fixed the size of the generated images, the lack of content edges in images of certain sizes was not used for metric calculation. StableDiffusion directly generates from noise at the target size, achieving a higher FID score, but slightly lower on CLIP-S and CLIP-A scores due to frequent stacking of repeated elements in the images. ElasticDiffusion's generated results lack clear content, making the images appear blurry, thus performing poorly on all metrics. SyncDiffusion

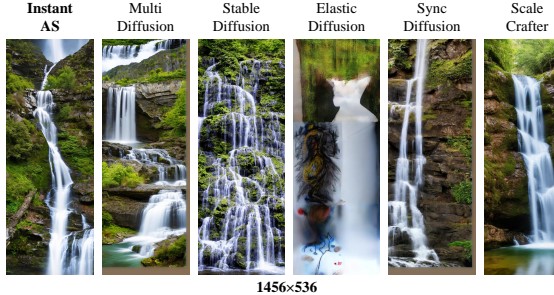

1456×536

*A photo of a cascading waterfall*

**Figure 6: Comparison of different methods in vertical image.**

**Table 2: Sampling Time and Memory Usage. We compare the sampling time and storage space required for all methods at five different sizes. The best-performing indicator is highlighted in bold, and the second-best indicator is underlined.**

| Resolution | Methods | Time | Memory Usage |
|---|---|---|---|
| 512 × 1024 | MultiDiffusion | 12.9s | 10.8GB |
| | StableDiffusion | 5.5s | 10.0GB |
| | ElasticDiffusion | 48.2s | 10.2GB |
| | SyncDiffusion | 93.8s | 23.7GB |
| | ScaleCrafter | 6.4s | 10.0GB |
| | **InstantAS** | **3.4s** | **9.8GB** |
| 512 × 2048 | MultiDiffusion | 27.5s | 12.9GB |
| | StableDiffusion | 11.5s | 12.8GB |
| | ElasticDiffusion | 54.7s | 12.9GB |
| | SyncDiffusion | 151.3s | 23.7GB |
| | ScaleCrafter | 16.7s | **10.0GB** |
| | **InstantAS** | **7.1s** | 12.8GB |
| 1024 × 512 | MultiDiffusion | 13.4s | 10.8GB |
| | StableDiffusion | 6.1s | 10.1GB |
| | ElasticDiffusion | 48.6s | **9.8GB** |
| | SyncDiffusion | 97.2s | 23.7GB |
| | ScaleCrafter | 6.7s | 10.0GB |
| | **InstantAS** | **3.8s** | 10.0GB |
| 2048 × 512 | MultiDiffusion | 28.2s | 12.9GB |
| | StableDiffusion | 13.2s | 12.6GB |
| | ElasticDiffusion | 55.8s | 16.8GB |
| | SyncDiffusion | 159.9s | 23.5GB |
| | ScaleCrafter | 17.1s | **10.0GB** |
| | **InstantAS** | **7.2s** | 12.9GB |
| 4096 × 4096 | MultiDiffusion | 4829.7s | 21.4GB |
| | StableDiffusion | - | - |
| | ElasticDiffusion | 172.8s | **17.8GB** |
| | SyncDiffusion | 30159.9s | 23.7GB |
| | ScaleCrafter | - | - |
| | **InstantAS** | **47.3s** | 19.2GB |

enhances the coordination consistency of images based on MultiDiffusion, so the overall performance metric is slightly higher than MultiDiffusion. ScaleCrafter performs quite well in most metrics, but the similar repeating elements among them result in a

slightly lower CLIP-A score. In contrast, InstantAS has good generation quality and diversity, along with good consistency with the prompt, showing a clear lead on all three metrics.

*Sampling Speed and Memory Usage Comparison.* We conducted another important test: Sampling Speed and Memory Usage, as shown in Table 2. It can be seen that ElasticDiffusion has a longer average sampling time. However, due to its characteristic of cropping and generating from fixed-size images, its sampling time is less affected by image size. The sampling time of MultiDiffusion increases significantly with the increase in image size and greatly exceeds the time taken by StableDiffusion. InstantAS far exceeds the other methods in sampling speed across all image sizes, with an average time that is only 1/4 of MultiDiffusion and even about half the average time of direct sampling in StableDiffusion, demonstrating the superiority of the non-overlapping minimum coverage sampling method. To further demonstrate the relationship between the time and memory required for various methods and image size, Table 2 also shows an extreme case: generating images of 4096 × 4096. It can be observed that the time required for MultiDiffusion sampling grows explosively with the increase in image size, reaching around an hour and a half. ElasticDiffusion, on the other hand, is less affected by image size. StableDiffusion and ScaleCrafter were unable to complete the sampling due to insufficient memory as it samples the entire image directly. The sampling speed of SyncDiffusion is the slowest among all methods. InstantAS maintains an extremely fast average sampling speed and reasonable range of memory usage, demonstrating strong stability.

## 4.3 Ablation Study

*Inter-Domain Distribution Bridging.* Table 3 demonstrates the impact of the Inter-Domain Distribution Bridging method on the experimental results by controlling variables. In this experiment, we fixed the size of the images at 512 × 2048 and sampled 700 images each time for metric calculation. Since the images of each area are generated separately without the Inter-Domain Distribution Bridging method, sampling takes less time, but the sampling results perform poorly in terms of image quality indicators. Figure 7 shows the impact of the proportion coefficient $\gamma$ on the generated results during domain distribution bridging. When $\gamma$ is small, the information fused in the sampling area during the subsequent generation step is also less compared to the previous generation step. Under the multi-step generation characteristic of the diffusion model, this allows the information from different areas to be merged more progressively and gradually. However, when $\gamma$ increases to 0.5, the result is similar to that of $\gamma$ at 1.0, indicating that when two identical sampling areas are divided into diffusion steps at a short period (with a period of 2 when $\gamma$ = 0.5, and a period of 1

**Table 3: Ablation Study**

| Method Details | FID↓ | CLIP-S↑ | CLIP-A↑ | Time↓ |
|---|---|---|---|---|
| Stable Diffusion (Baseline) | 23.17 | 0.21 | 5.45 | 12.9s |
| w/o *Non-Overlapping* | 24.34 | **0.24** | 6.28 | 45.2s |
| w/o *Distribution Bridging* | 103.62 | 0.09 | 3.02 | **7.0s** |
| **Complete InstantAS** | **22.66** | 0.23 | **6.39** | 7.6s |

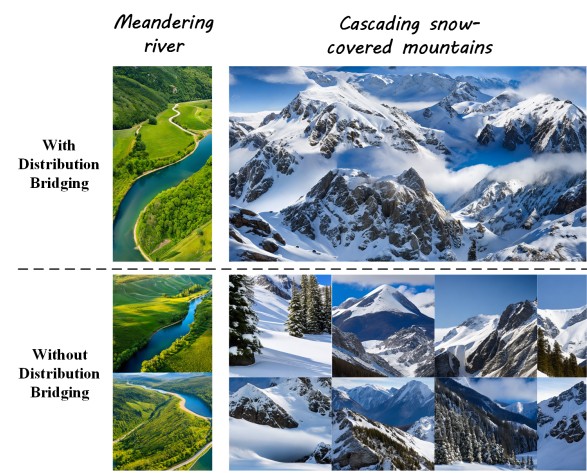

$\gamma = 0.015$ (Period=200)   $\gamma = 0.125$ (Period=8)   $\gamma = 0.25$ (Period=4)   $\gamma = 0.50$ (Period=2)   $\gamma = 1.00$ (Period=1)

**Figure 7: The generated results under different $\gamma$. When the setting of $\gamma$ shortens the period at which two identical sampling regions are partitioned, the boundaries become more pronounced. However, when the period length exceeds the number of sampling steps, continuing to lengthen the period will not have a major impact on the results. Therefore, we set $\gamma$ to 0.015.**

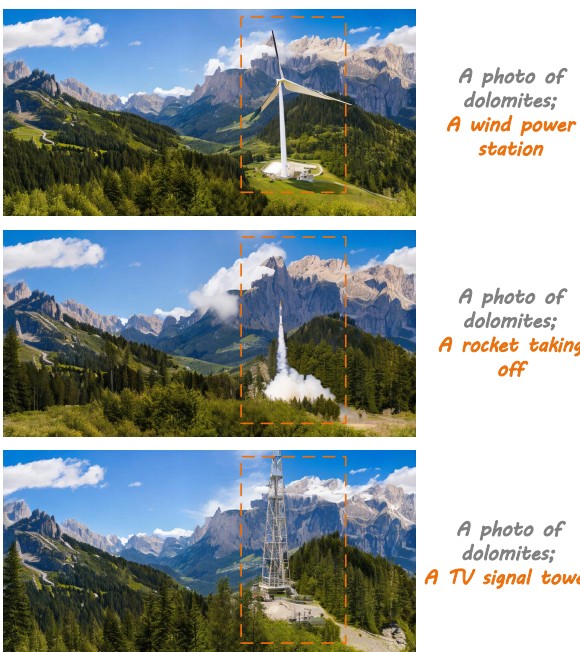

**Figure 8: Ablation study of Inter-Domain Distribution Bridging method. There are obvious splicing gaps between the sampling regions without the inter-domain distribution bridging method. However, by using this method, the blending between sampling areas becomes relatively smooth.**

**Figure 9: Experimental results of region-controlled generation. The size of the image is $540 \times 1082$.**

when $\gamma = 1.0$), the fusion effect is worse. Figure 8 shows the impact of using the Inter-Domain Distribution Bridging method on the generated results. When each region is generated separately, their generation paths disperse unconstrainedly, showing clear edges. However, after the application of the Inter-Domain Distribution Bridging method, the information from different regions blends with each other, making their transitions more natural.

*Non-Overlapping Minimum Coverage.* Table 3 demonstrates the impact of Non-Overlapping Minimum Coverage method on the experimental results. As a comparison,, we overlapped adjacent two sampling regions by 90%, and at the end of each sampling step, the average feature maps of all overlapping regions were calculated as the initial values for the next sampling step, as described in Figure 3. The Non-Overlapping method can significantly reduce the sampling time while maintaining the image quality.

### 4.4 Region Control Generation

In Figure 9, we demonstrate the effect of region control generation. In this experiment, we give different foregrounds (orange) to the same semantically controlled region division while retaining the background (gray). Experimental results suggest that the region control method achieves a good fusion and control effect.

## 5 CONCLUSION

In this paper, we propose a method for generating images of arbitrary sizes: InstantAS, which is used for pre-trained text-to-image diffusion models and requires no additional training. The highlight of InstantAS is the non-overlapping division of the sampling region, which effectively reduces the information redundancy in image sampling of any size and greatly increases the sampling speed. The inter-domain distribution bridging method effectively prevents obvious splicing gaps caused by non-overlapping divisions and achieves high-quality global fusion. In addition, we also explored the method of partitioned region control generation, using dynamic guidance cross-attention to dynamically adjust the guidance information in different regions. However, our method still has some shortcomings, such as lack of global information. We will present and discuss these shortcomings in detail in the supplementary material. InstantAS reveals the potential of partitioned area generation in the task of generating images of arbitrary sizes. In future work, we will further explore methods that generate images while maintaining resolution, balancing local and global information.

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
