# OpenReview forum: "InstantAS: Minimum Coverage Sampling for Arbitrary-Size Image Generation"
_acmmm.org/ACMMM/2024/Conference — MM2024 Poster_

### Official Review · Reviewer_XC9P · 2024-05-06

**Rating:** 2
**Confidence:** 3

**Summary:**

This paper aims to tackle the task of arbitrary-size text-to-image generation.
To this end, this paper proposes InstantAS, which enables the pre-trained text-to-image generation models to synthesize images with resolutions out of the training distribution without additional training.
Specifically, this paper utilizes Non-Overlapping Minimum Coverage Sampling and Inter-Domain Distribution Bridging to generate patch-by-patch and fuse them into a panorama.
In addition, InstantAS further extends the method to region control generation, that is, it could synthesize images of arbitrary resolution according to different text prompts in various regions.
Experiments have demonstrated the performance and efficiency advantages of the proposed method on arbitrary resolution image generation tasks.

**Strengths:**

1. The paper is well-written and easy to follow.

2. Extending the existing pre-trained text-to-image generation models towards arbitrary-resolution generation is a worthy direction to explore, and this is in line with the topic of Multimedia.

3. The proposed method demonstrates excellent performance on multiple metrics in quantitative evaluation under several settings, especially, it can synthesize higher-resolution images with lower computational overhead and inference time.

**Limitations:**

1. The conclusion lacks essential support: In the abstract, it is claimed that the proposed 'InstantAS can be applied to nearly any existing pre-trained Text-to-Image diffusion models, however, the method and experiments in the paper only illustrate the effect of InstantAS on Stable Diffusion. As a result, the proposed method lacks universal verification and the conclusion lacks strong support.

2. Lack of description of the experiment details: The paper lacks specific description of the dataset used, such as the input prompts and reference images information used to calculate metrics such as FID.

3. Limited novelty: The proposed method is like a non-overlapping extension of overlapping sampling in MultiDiffusion; region-controlled generation has also been relatively mature in ReCo [1*], R&B [2*] and other methods, and combining arbitrary resolution generation with them is not an innovative contribution point.

4. Quantitative results: Although the proposed method can achieve ideal results in metrics such as FID and CLIP-S under multiple settings, it is not stable and the improvement of the method is very limited.

5. Qualitative results: Multiple visualizations of the proposed InstantAS have obvious inconsistencies between patches, which are instead not obvious in other methods such as MultiDiffusion and StableDiffusion. Moreover, there also exist obvious object repetition in the generated results of InstantAS, which makes the qualitative superiority of the proposed method less convincing.

[1*] ReCo: Region-Controlled Text-to-Image Generation

[2*] R&B: Region and Boundary Aware Zero-shot Grounded Text-to-image Generation

**Suitability:**

3

---

### Official Review · Reviewer_1yQ5 · 2024-05-16

**Rating:** 4
**Confidence:** 2

**Summary:**

This paper introduces the InstantAS method for generating images of arbitrary sizes. The method improves sampling speed and image consistency through minimum coverage segmentation and cross-domain distribution integration, and employs a dynamic semantic-guided cross-attention approach to control generation in different regions. The research problem addressed in the paper is highly practical, and the proposed method is closely tied to the problem, with innovative theoretical foundations. Additionally, the authors present extensive experimental results, discussing the generation performance under different conditions in a thorough and reasonable manner.

**Strengths:**

1. The paper has a clear structure, with concise and easy-to-understand writing, and provides a comprehensive summary of related work in the research field.
2. The research problem is well-defined: the generation of text-to-image at arbitrary sizes, with strong practical value and simple, effective methods.
3. The proposed method effectively addresses the problem of generating images at arbitrary sizes, resulting in natural images. Section 3.3, "Dynamic Semantic Guided Cross-Attention," shows some theoretical innovation, though limited, as adjusting Attention in U-Net has been common in earlier works.
4. The experiments are thorough, discussing the advantages of the method in visual quality, time complexity, and quantitative comparisons under different metrics. The comparative experiments are sufficient and demonstrate the effectiveness of the method convincingly.

**Limitations:**

1. It is expected that the author's team will further open-source the work presented in the paper.
2. There is an extra comma in line 857 of the left column after "As a comparison."
3. From a theoretical contribution perspective, Section 3.1, "Non-Overlapping Minimum Coverage Sampling," can optimize the speed of generating arbitrary sizes in the Diffusion Model, but I lean towards considering it as an engineering technique.
4. Section 3.2, "Inter-Domain Distribution Bridging," can similarly be viewed as an engineering technique; it is suggested that the authors better highlight the theoretical contributions of the paper.

**Suitability:**

2

---

### Official Review · Reviewer_QMcu · 2024-05-24

**Rating:** 4
**Confidence:** 3

**Summary:**

This paper proposes a generation method that does not require massive overlapping sampling for the task of generating images of arbitrary size. The proposed method achieves inter-region communication through region reorganization and inter-domain distribution bridging. Benefiting from not requiring intensive overlapping samples, the approach reduces computational consumption.

**Strengths:**

- The methodology presented in this paper is well motivated and the proposed methodology is intuitive and reasonable.
- The proposed method is training-free and the results presented in the paper show good generation quality.
- The paper is well-written and easy to understand.

**Limitations:**

My main concern with this method is that the method presented in the paper may be difficult to apply to complex content, such as generation involving multiple objects or concepts (e.g. "A dog standing beside a car"). The generated results shown in this paper are all simple landscapes, and the difficulty of generating such images is relatively low.

**Suitability:**

3

---

### Meta-Review · Area_Chair_ehZQ · 2024-07-02

**Recommendation:** Accept (Poster)
**Confidence:** 4

**Metareview:**

All the reviewers appreciate that this work studies an interesting problem of text-to-image generation with arbitrary sizes. The technical solution is efficient with practical values in visual performance and running time. The rebuttal has largely addressed the concerns of the reviewers, and two reviewers raised their ratings. In the end, the final ratings are Weak Accept, Borderline Accept, and Borderline Reject. The authors should reconsider the claim that "the method is competent for almost all diffusion models." The claim is better revised or removed. The AC recommends the acceptance of this paper.